# MAGI1, a Scaffold Protein with Tumor Suppressive and Vascular Functions

**DOI:** 10.3390/cells10061494

**Published:** 2021-06-14

**Authors:** Janine Wörthmüller, Curzio Rüegg

**Affiliations:** Laboratory of Experimental and Translational Oncology, Pathology, Department of Oncology, Microbiology and Immunology (OMI), Faculty of Science and Medicine, University of Fribourg, CH-1700 Fribourg, Switzerland

**Keywords:** breast cancer, tumor suppressor, PTEN, PI3K/AKT, Wnt, focal adhesions, integrins, inflammation, NSAIDs, prognostic marker

## Abstract

MAGI1 is a cytoplasmic scaffolding protein initially identified as a component of cell-to-cell contacts stabilizing cadherin-mediated cell–cell adhesion in epithelial and endothelial cells. Clinical-pathological and experimental evidence indicates that MAGI1 expression is decreased in some inflammatory diseases, and also in several cancers, including hepatocellular carcinoma, colorectal, cervical, breast, brain, and gastric cancers and appears to act as a tumor suppressor, modulating the activity of oncogenic pathways such as the PI3K/AKT and the Wnt/β-catenin pathways. Genomic mutations and other mechanisms such as mechanical stress or inflammation have been described to regulate MAGI1 expression. Intriguingly, in breast and colorectal cancers, MAGI1 expression is induced by non-steroidal anti-inflammatory drugs (NSAIDs), suggesting a role in mediating the tumor suppressive activity of NSAIDs. More recently, MAGI1 was found to localize at mature focal adhesion and to regulate integrin-mediated adhesion and signaling in endothelial cells. Here, we review MAGI1′s role as scaffolding protein, recent developments in the understanding of MAGI1 function as tumor suppressor gene, its role in endothelial cells and its implication in cancer and vascular biology. We also discuss outstanding questions about its regulation and potential translational implications in oncology.

## 1. Introduction

MAGI1 (membrane-associated guanylate kinase, WW and PDZ domain-containing protein 1), discovered in 1998 and named BAP1 (brain-specific angiogenesis inhibitor 1-associated protein) [1], was initially described as an intracellular adaptor protein involved in stabilizing epithelial junctions [2]. Located on chromosome 3p14.1, MAGI1 is the only known protein in which a GUK domain is combined with five PDZ domains and two WW domains [3]. Through these domains, MAGI1 recruits a number of different kinds of molecules to strengthen the junctional complex. Since its initial discovery, a growing number of studies have point towards a more relevant role for MAGI1 than being a simple scaffold protein. MAGI1 has been described to be involved in multiple, complex, and essential cellular functions, like in regulating cell–cell and cell–matrix adhesion, but also in mediating cell migration, signaling, proliferation, and survival. In specialized polarized tissues, such as the nervous system and the kidney, MAGI1 is involved in receptor synaptic localization and the homeostasis of ion channels [4]. The most intriguing emerging roles of MAGI1 are probably related to its function in vascular biology and its role in cancer as tumor suppressor. Endothelial MAGI1 regulates important vascular functions, such as vascular integrity and permeability, response to shear stress and nitric oxide (NO) production, and angiogenesis. MAGI1 acts as tumor suppressors in a multitude of cancers, in particular liver, colorectal, cervical, breast, brain, and gastric cancers, serving in some cases as a prognostic marker. MAGI1 participates in regulating oncogenic pathways such as the PI3K/AKT and the Wnt/β-catenin pathways, however, the mechanisms of its downregulation in cancer remain mostly elusive. Several studies have found that in tumors, *MAGI1* undergoes mutations, gene rearrangements, and methylation. MAGI1 is also regulated post-transcriptionally by miRNAs, or at the protein level by degradation or phosphorylation. Other mechanisms such as mechanical stress or inflammation also regulate MAGI1 expression. This is an exciting journey for a gene that has not yet entered the full spotlight (since its discovery, there are only about 300 references about MAGI1 in PubMed; compared to the nearly over 80,000 references for VEGF, 100,000 for p53 during the same period, and over 85,000 for Sars-Cov-2 in the last two years).

Here, we summarize some of the most important findings about MAGI1, in particular concerning its emerging role as vascular regulator and tumor suppressor, along with the mechanisms that regulate its expression. We end up listing a number of open questions that have no clear answers yet, and that we believe deserve further attention.

### 1.1. MAGI1 Structure and Expression

MAGI1 is a member of the large family of MAGUK (membrane-associated guanylate kinases) scaffold proteins, that play essential roles during development, cell–cell communication, and cellular signal transduction. Some of their diverse functions include the regulation of cellular processes such as cell polarity, tight junction (TJ) formation, neuronal synaptic transmission, cell proliferation, and apoptosis. Accordingly, mutations or changes in their expression are linked to defects in cell–cell adhesion, cell polarity, cell proliferation, and development. In mammals, the MAGUK family englobe 22 members classified phylogenetically into 8 sub-families, which vary in size, domain organization, localization, and biological functions. Despite their differences, all the MAGUKs share a well conserved core structure, which is comprised of one or multiple PDZ domains (except for CACNB) in the amino-terminal region, a catalytically inactive guanylate kinase (GUK) domain and a Src Homology 3 (SH3) domain (except for the MAGI subfamily). In contrast to other members of this family, MAGI1 has some distinct features: (1) the GUK domain is located in the amino-terminal region rather than at the carboxyl end; (2) its SH3 domain is replaced by two WW domains situated downstream of the GUK domain (that function in a similar fashion as the SH3 domain), and (3) it contains five PDZ domains instead of the usual one or three (reviewed in [3,5,6]). The name PDZ domain is derived from three members of the MAGUK family: PSD-95, DLG, and ZO-1 [7], however, PDZ domains are not restricted to MAGUKs. The PDZ domains are modules of 80–90 amino acids that have been found either in single or multiple copies within a broad range of proteins, and in diverse organisms such as mammals, yeasts, or plants [8]. This domain is one of the most prevalent protein–protein interaction modules in multicellular eukaryotic genomes [9], and the majority of proteins containing it are associated with the plasma membrane and restricted to specific sub-cellular domains, such as synapses, cell–cell junctions, apical or lateral plasma membranes [10]. Regarding the GUK domain, it shares approximately 40% sequence identity to the yeast guanylate kinase, however, it shows very weak binding affinity to GMP and ATP; functional studies indicate that none of the GUK domains in MAGUK proteins are catalytically active [11]. Nevertheless, it appears that they have maintained the guanylate kinase structure to mediate protein–protein interactions independently of its predicted enzymatic activity [10].

All MAGUKs localize to regions of cell–cell contact, such as TJs in epithelial cells and synaptic junctions in neurons, where they orchestrate the assembly of multiprotein complexes via their protein–protein interaction domains [12,13]. The MAGI subfamily contains MAGI1, MAGI2, and MAGI3 and they are present in the TJs of epithelial cells [3]. MAGI2 was initially identified in rat, as a protein interacting with *N*-methyl-d-aspartate receptors (NMDARs) and neuronal cell adhesion proteins, and was termed S-SCAM [14], while MAGI3 (also named SLIPR) was identified in a two-hybrid screening as a protein interacting with the tumor suppressor phosphatase and tensin homolog (PTEN) [15] and the receptor tyrosine phosphatase β (RPTPβ) [16]. There is a high homology among the three MAGI proteins, especially regarding their structural domains.

MAGI1 was first identified in the mouse, as a protein interacting with K-RasB. MAGI1 mRNA undergoes alternative splicing to produce three transcripts that encode proteins with a unique carboxyl-terminal region, named MAGI1-a, MAGI1-b, MAGI1-c. These different isoforms differ in length, cellular localization, and tissue distribution. While MAGI1-a is the shortest variant, encoding for a protein of 1139 amino acids, MAGI1-b and MAGI1-c encode for proteins of 1171 and 1374 amino acids, respectively. In addition, the carboxyl-terminal region of MAGI1-c contains 251 amino acids forming a three bipartite nuclear localization signal (NLS) (reviewed in [3]). Regarding their tissue distribution, MAGI1 is widely expressed, with the exception of skeletal muscle. The variants MAGI1-a and MAGI1-b are present in non-epithelial tissues; MAGI1-a is highly expressed in the brain and pancreas, while MAGI1-b is predominantly expressed in the brain and heart. On the contrary, MAGI1-c is present predominantly in epithelial tissues, such as the colon, kidney, lung, liver, and pancreas. In addition, all three MAGI1 isoforms are also present in neuronal tissue [17].

### 1.2. MAGI1 Role as a Scaffold Molecule

In epithelial cells, MAGI1 mostly localizes at TJs and adherens junctions, where it interacts with various molecules functioning as a scaffold protein [2,17]. Multi-cellular organisms are separated from the external environment by a layer of epithelial cells whose integrity is maintained by intercellular junctional complexes. Epithelial junctions is a broad term that includes tight, adherent, and gap junctions and desmosomes [18]. TJs are essential in the physiology of epithelial cells; they constitute not only a semipermeable barrier that regulates the flow of ions, solutes, and cells across paracellular spaces, but they also contribute to the establishment and maintenance of apico-basal polarity, and they are targets and effectors of signaling pathways controlling gene expression, cell differentiation, and proliferation processes (reviewed in [19]).

MAGI1 has been described to interact with different classes of proteins, including integral membrane proteins or TJ-associated scaffolding proteins such as ZO-1 (another member of the MAGUK family) [17], JAM4 [20], and ESAM [21]; and TJ-associated signaling molecules like K-RAS [3], Rho family guanine nucleotide exchange factor (GEF) termed mNET1 [22], Rap GDP/GTP exchange protein (GEP) [23], β-catenin [24], RPTPβ [16], or the tumor suppressor PTEN [25]. MAGI1 colocalizes with β-catenin, E-cadherin [26], and PTEN in basolateral adherens junctions, where it regulates the formation and maintenance of E-cadherin containing complexes [25]. Recently, TRIP6 was identified as a new scaffold molecule binding MAGI1 and forming part of the MAGI1/PTEN signalosome, where it plays a critical role in the maintenance of cell–cell contacts and the control of invasiveness [27]. MEK1 (MAPK/ERK kinase 1) is also necessary for PTEN membrane recruitment, being MAGI1 the scaffold mediating this interaction [28]. Additionally, MAGI1 has been described to bind to the adenoviral E4-ORF1, the high-risk HPV E6 oncoproteins [29], and to Tax1, an oncoprotein encoded by the human T-cell leukemia virus type 1 (HTLV-1), the etiological agent of adult T-cell leukemia (ATL) [30].

MAGI1 also binds to synaptopodin and α-actinin-4 in glomerular podocytes through its second WW and fifth PDZ domain, respectively [31]. These findings suggested early on that MAGI1 plays a role in actin cytoskeleton dynamics within polarized epithelial cells. Likewise, MAGI1 colocalizes with megalin, a transmembrane endocytic receptor glycoprotein, in podocytes [32]. MASCOT/AMOTL2 was also identified as a binding partner of MAGI1 interacting with its WW domain binding motif [33]. MAGI1 also binds to nephrin, a transmembrane component of the slit diaphragm [34].

MAGI1, 2, and 3 have also been described to interact with a diversity of other ligands from the nervous system, including NMDA receptors, neuroligin, BAI1, activin type II receptor, δ-catenin, the muscle-specific tyrosine kinase (MuSK) (reviewed in [17]), atrophin-1 [35], and the acid-sensing ion channel-3 (ACCN3) in neurons [36]. MAGI3 was also found to bind to frizzled-4, -7, and Ltap, forming a ternary complex involved in regulating the JNK signaling pathway [37]. MAGI2 also interacts with Smad3, a TGF-β downstream molecule [38], and similarly to MAGI1 and MAGI3, it also interacts with PTEN [39]. Figure 1 summarizes some of the most relevant binding partners of MAGI1 (for a more comprehensive overview on MAGI interacting molecules, we refer to excellent review articles [19,40,41]).

### 1.3. MAGI1 Role as Tumor Suppressor

Due to the critical role of MAGI1 in regulating cell–cell contacts, it is not surprising that its loss influences the adhesion, proliferation, invasiveness, and metastasis of different tumor cell types. Disruption of cadherin junctional complexes is the hallmark of cancer progression and has been extensively associated with invasiveness and metastasis [27]. The stability of these complexes is under the control of molecular scaffolds and several signaling pathways, including the PTEN tumor suppressor, that interacts indirectly with β-catenin by binding to MAGI1-b [47]. PTEN is a lipid phosphatase that antagonizes the action of phosphoinositide 3-kinase (PI3K) signaling—which regulates growth, survival, and proliferation—by dephosphorylating the lipid signaling intermediate phosphatidylinositol-3,4,5-trisphosphate (PIP_3_) [48]. Kotelevets et al. demonstrated that ectopic expression of MAGI1-b potentiated the interaction of PTEN with junctional complexes, promoting E-cadherin-dependent cell–cell aggregation, and reverting the Src-induced invasiveness of kidney epithelial MDCK*ts-src* cells [25]. In addition, MAGI1-b was shown to slightly decrease the activity of AKT, also known as protein kinase B (PKB), a downstream effector of PI3K and, together with PTEN, to stabilize adherens junctions and suppress the invasiveness of these cells. Taken together, MAGI1 seems an important molecule for the stabilization of cadherin mediated cell–cell interactions and the suppression of invasiveness in non-transformed epithelial cells [4].

MAGI1 is downregulated in various cancers and acts as a tumor suppressor; in several cases by modulating direct or indirectly PTEN activity. For example, Zhang et al. demonstrated that transfection of MAGI1 in HepG2 cells inhibits cell migration and invasion by upregulating the expression of PTEN in hepatocellular carcinoma (HCC) [49]. In addition, decreased expression of MAGI1 was found to be associated with poor prognosis of HCC, its downregulation correlated with vascular invasion of HCC tissues [50]. In T-cells, MAGI1 has been shown to inhibit AKT activity through its interaction with PTEN and MEK1. In the study of Kozakai et al., they found that MAGI1 expression is decreased in multiple human T-cell leukemia types, including adult T-cell leukemia (ATL). Whereas the knockdown of MAGI1 increases AKT and MEK activities, overexpression of MAGI1 in a MAGI1-low ATL cell line reduces cellular growth and AKT and MEK signaling [51].

Other MAGI family members have also been described to regulate cell migration and invasion via PTEN modulation. MAGI2 was found to inhibit cell migration and proliferation in different HCC cell lines by diminishing the phosphorylation of FAK and AKT [52], two important signaling molecules promoting cell migration and proliferation [53]. Likewise, the studies of Ma et al. in glioma cells showed that overexpression of MAGI3 upregulates PTEN protein expression, inhibits the phosphorylation of AKT, and suppresses cell proliferation [54]. In a study in lung adenocarcinoma, different lung adenocarcinoma cell lines stimulated with TGF-β1 induced the overexpression of the microRNAs miR-134/487b/655, which in turn targeted MAGI2 for silencing. As a consequence, PTEN activity is reduced, resulting in TGF-β-induced epithelial–mesenchymal transition (EMT) followed by acquired resistance to epidermal growth factor receptor (EGFR) tyrosine kinase inhibitors (TKI) [55]. A role of MAGI2 in prostate tumorigenesis has been also evaluated; mutations in *MAGI2* contribute to prostate carcinogenesis by driving AKT phosphorylation and altering PI3K signaling, thus disrupting PTEN signaling [56].

MAGI has been also found to modulate other signaling pathways such as the Wnt/β-catenin, the PI3K/AKT or the mitogen-activated protein kinase (MAPK)/extracellular signal-regulated kinase (ERK) signaling pathways. Three studies performed in glioma cells showed similar outcomes; the study of Lu et al. demonstrated that MAGI1 played an essential role during glioma progression; its silencing in different glioma cell lines enhanced proliferation and inhibited apoptosis, increased Wnt/β-catenin signaling, enhanced AKT phosphorylation, and reduced E-cadherin and PTEN expression. Conversely, overexpression of MAGI1 significantly inhibited tumor growth in vivo [57]. Another study found that overexpressing MAGI1 inhibits proliferation, migration, and invasion of glioma cells by regulating cell growth and EMT through AKT, matrix metalloproteinase 2 (MMP2) and MMP9, and the E-cadherin/*N*-cadherin/vimentin pathway [58]. MAGI3 was also shown to negatively regulate the Wnt/β-catenin signaling suppressing the malignant phenotypes of glioma cells. Experimentally, MAGI3 knockdown enhanced cell proliferation in vitro, while MAGI3 overexpression suppressed tumor growth in vivo. Moreover, MAGI3 expression levels were found to be negatively associated with tumor grade and poor prognosis and activation of Wnt/β-catenin signaling in glioma datasets [59]. Likewise, in a study of Zaric et al., MAGI1 was identified as a negative regulator of the WNT/β-catenin signaling pathway, with tumor-suppressive and anti-metastatic activity in colorectal cancer (CRC). In this case, MAGI1 silencing was shown to decrease E-cadherin and β-catenin localization at cell–cell junctions; enhancing Wnt signaling, migration, and invasion in vitro. Conversely, overexpression of MAGI1 suppressed cell migration and invasion in vitro in different CRC cell lines and attenuated primary tumor growth and spontaneous lung metastasis in vivo in an orthotopic model of CRC by inhibiting the Wnt/β-catenin signaling pathway [60].

Recently, MAGI1 was reported to be expressed at higher levels in estrogen receptor (ER)^+^/HER2^-^ breast cancers (BCs), relative to HER2^+^ and triple negative (TN) BCs. More specifically, within this subset, high MAGI1 expression was associated with better prognosis, while low MAGI1 levels correlated with higher histological grade, more aggressive phenotype, inflammation, and worse prognosis. Experimentally, MAGI1 downregulation in ER^+^ BC cell lines promoted cell proliferation and survival in vitro and enhanced primary tumor growth and lung metastasis formation in vivo [61]. MAGI1 downregulation caused the activation of the PI3K/AKT and Wnt signaling pathways, suggesting their involvement in these cellular effects. A similar study that also evaluated the expression of MAGI1 in BC cell lines confirmed its tumor suppressive function and found a distinct molecular mechanism behind this effect. The study reported that loss of MAGI1 leads to cellular accumulation of E-cadherin and the actin binding scaffold AMOTL2, resulting in increased stiffness, which in turn downregulates yes-associated protein 1 (YAP) activity, the terminal Hippo-pathway effector, and increases rho-associated protein kinase (ROCK) and p38 stress activated protein kinase activities [62].

Jia et al. showed that MAGI1 downregulation promoted the activation of pathways involved in metastasis and EMT in gastric cancer cells. In a panel of different gastric cancer cell lines, MAGI1 knockdown significantly enhanced migration and invasion of these cells by altering the expression of MMPs and EMT-related molecules via inhibiting the MAPK/ERK signaling pathway [63]. 

MAGI1 was also shown to be regulated by oncogenic viruses, playing an important role in the consequent neoplastic transformation of the cells. There is increasing evidence of a link between the tumorigenic potential of certain viruses and its capacity to functionally inactivate certain TJ-associated proteins [64]. Notably, loss of cell–cell adhesion and cell polarity are common features of epithelial-derived cancer cells [65]. In the context of cervical cancers, MAGI1 is a sensitive proteolytic substrate for the human papillomavirus (HPV)-16 and HPV-18 E6 oncoproteins [66]. The study of Kranjec et al. showed that restoration of MAGI1 expression by introducing a mutant MAGI1 resistant to E6-targeting and subsequent degradation in HPV-positive tumor cells, resulted in the induction of apoptosis and repression of cell proliferation [67]. A common mechanism by which the oncoproteins of DNA tumor viruses promote tumorigenesis is by inactivating tumor suppressor proteins. In this regard, E6 oncoproteins target MAGI1 for proteasome-mediated degradation, whereas 9ORF1, another known oncoprotein that targets MAGI1, functions sequestering it in the cytoplasm, preventing its localization at the plasma membrane at sites of cell–cell contacts [29]. Table 1 summarizes the tumor suppressor functions of MAGI in different cancers.

### 1.4. MAGI1 Vascular Functions

While significant insights have been made in confirming the critical role of MAGI1 as tumor suppressor in different cancer types, research involving MAGI1 functions in endothelial cells (ECs) remains limited. The vascular endothelium, located at the interface between tissues and blood, plays a crucial role in the regulation of vascular permeability, thrombus formation, oxidative stress, and angiogenesis [68]. These functions largely depend on the interaction between ECs, that maintain the vascular barrier integrity by establishing a set of intercellular adhesion, such as TJs and adherens junctions [69], and their interaction with the surrounding extracellular matrix (ECM), through integrin-dependent adhesions and focal complex formation. These adhesive complexes not only provide an anchor point for ECs to adhere to the substratum, but they also transmit forces and biochemical signals between the cells and the matrix [70]. It is well known that ECs dysfunction triggers the initiation and progression of a wide range of vascular diseases [71].

A first indication that MAGI1 may be involved in regulating angiogenesis was suggested when MAGI1 was cloned in 1998 and characterized as a protein interacting with brain-specific angiogenesis inhibitor 1 (BAI1) [1], an adhesion G protein-coupled receptor with anti-angiogenic and anti-tumorigenic properties [72], whose expression decreases during glioma formation. Moreover, BAI1 expression was also shown to inhibit angiogenesis and metastasis in colorectal cancer and stromal vascularization in pulmonary adenocarcinoma [73,74]. Likewise, MAGI1 was shown to bind to PRKX, a protein kinase that modulates angiogenesis, however, if MAGI1 regulates PRKX functions through this interaction needs further evaluation** [75]. A recent study showed that MAGI1 downregulation in ECs promotes pseudocapillary and endothelial network formation, while MAGI1 overexpression reduces endothelial cell tubulogenesis in vitro and angiogenesis in vivo in a transgenic mouse model [76].

Additional studies have pointed towards a role of MAGI1 in regulating ECs adhesion, their strength and dynamics, and eventually angiogenesis. The study of Sakurai et al. reported that MAGI1 mediated Rap1 activation (by the direct binding of MAGI1 to the guanine nucleotide exchange factor for Rap1, PDZ-GEF1), and enhanced vascular endothelial (VE)-cadherin-dependent endothelial cell–cell adhesion, while MAGI1 depletion impaired it [43]. Similarly, MAGI1 was shown to promote and strengthen mature cell–cell adhesion via activation of RhoA by binding to ESAM [44], resulting in MAGI1 recruitment to intercellular contacts through the PDZ-binding domain. In addition, MAGI1 was found to localize at cell contacts between human umbilical vein endothelial cells (HUVECs) and mouse ECs. Localization studies in transfected Chinese hamster ovary (CHO) cells revealed that MAGI1 was only recruited to cell contacts if ESAM was co-expressed [21], and co-localization of both proteins in these structures promoted actin polymerization and activated RhoA [44]. Recently, MAGI1 was found present in mature focal adhesions in HUVEC cells in colocalization with the known focal adhesion-associated molecules paxillin, α-actinin and β3-integrin. Experimental downregulation of MAGI1 in HUVEC reduced focal adhesion formation and maturation, cell spreading, actin stress fiber formation, and RhoA/Rac activation [76]. This is consistent with MAGI1 contributing to regulate focal adhesion maturation. Taken together, these results indicate that, at least in ECs, MAGI1 regulates focal adhesion dynamics by promoting their maturation and preventing their turnover, resulting in stabilized cell adhesion, decreased migration, and reduced angiogenesis.

MAGI1 has been also reported to regulate endothelial nitric oxide synthase (eNOS) expression and vascular nitric oxide (NO) production through the cyclic AMP-dependent protein kinase/AMP-activated protein kinase (PKA/AMPK) signaling pathway in response to fluid shear stress [77], the frictional forces generated by the blood flow on the luminal surface of the wall [78]. NO is a key regulator of homeostasis and adaptive responses of the vascular system [79]. Reduction in its production is associated with endothelial dysfunctions such as in hypertension, diabetes, or atherosclerosis [80]. In this context, a recent study found that proatherogenic agonists, which include inflammatory cytokines and growth factors as well as disturbed flow, cause EC activation. In ECs and macrophages from atherosclerotic-prone regions of mouse, MAGI1 expression was found to be upregulated. Furthermore, MAGI1 was shown to regulate NF-κB activation in ECs induced by various proinflammatory stimuli and to regulate endoplasmic reticulum stress-induced apoptosis, a critical molecular event for atherogenesis, via posttranslational modifications [81]. More specifically, p90RSK was shown to phosphorylate MAGI1 S741, causing MAGI1 de-SUMOylation at K931, thus leading to EC activation. In addition to its role as regulator of EC activation, MAGI1-large tumor suppressor kinases (LATS) 1/2-YAP interaction was shown to modulate endothelial permeability. Couzens and Weiss et al. found that MAGI1 is one of the binding partners of the LATS1 [82], and Abe et al. discovered that MAGI1 inhibits LATS1/2 expression, maintaining the EC barrier function by regulating the Hippo pathway [83]. In their study, they show that MAGI1 regulates LATS1 and 2 expression transcriptionally, and MAGI-mediated reduction of LATS1 and 2 expression leads to increased YAP activation, which is critically involved in the maintenance of EC barrier function (i.e., inhibition of EC permeability). Altogether, these data suggest a crucial role of the p90RSK-MAGI1 pathway in promoting both EC activation and permeability. This is a critical effect as the coordination of adhesion molecule expression with EC barrier opening is necessary for leukocyte and monocyte transendothelial migration and extravasation during acute inflammation.

Different components of cell junctions, including some direct MAGI1 interacting partners, have been described to modulate ECs permeability and leukocyte extravasation. Under physiological conditions, the endothelium is characterized by low permeability to solutes and leukocytes [84]. Under conditions of tissue damage, acute inflammation and vascular injury, there is an increase in vascular permeability for plasma accompanied by increased adhesion of leukocytes to the endothelium and extravasation [85]. Importantly, increased vascular permeability and leukocyte extravasation is promoted by the loosening of EC-EC contacts [86], following tyrosine phosphorylation of VE-cadherin, which can be induced by several factors, in particular vascular endothelial growth factor (VEGF) [87]. VE-cadherin is required for the recruitment of MAGI1, and together with Rap1, they finely modulate endothelial responses and barrier function [43,85]. Different JAM subfamily members have also been described to regulate leukocyte trafficking during inflammation and angiogenesis [88]. JAM4 was originally characterized as an adhesion molecule that regulates permeability of epithelial cell monolayers in association with MAGI1 [20]. Likewise, ESAM was shown to regulate vascular permeability in mouse models of inflammation [89]. In addition, ESAM supports the activation of the GTP-ase Rho, known to be involved in leukocyte transendothelial migration. A range of different EC surface proteins from the EC junctions are involved in the process of leukocyte extravasation (for more information see [90,91]). Taken together, being MAGI1 an important component of cell junctions, and considering its interaction with different adhesion molecules directly implicated in regulating leukocyte extravasation, we hypothesize that MAGI1 could play an active role in this process. This hypothesis, however, awaits experimental confirmation.

### 1.5. MAGI1 Function in Podocytes and the Nervous System

The podocyte of kidney glomeruli shares many cell biological characteristics with neuronal dendrites: both are postmitotic, polarized, and highly ramified cell types. They also share the expression of various molecules working for signal transduction, transmembranous transport, and intercellular contacts. They display the same cytoskeletal organization, with actin and some expression-restricted actin-binding molecules, such as synaptopodin, localized in podocyte foot processes and neuronal dendritic spines. In addition, podocytes express a wide variety of neurotransmitter receptors (reviewed in [92,93]). Evidence shows that MAGI1 is involved in receptor synaptic localization and the homeostasis of ion channels in both, the nervous system and in podocytes (reviewed in [4]).

At the glomerulus, MAGI1 is present in podocytes at the slit diaphragm (SD), the structure responsible for the glomerular filtration barrier that prevents serum proteins from leaking into the urine [94]. There, MAGI1 interacts with sidekick-1, a protein upregulated in podocytes of patients with focal segmental glomerulosclerosis [95]. In the retina, all three MAGI subfamily members are expressed by subsets of retinal neurons, there, sidekick interaction with MAGI proteins is necessary for the localization and the developmental function of this synaptic adhesion molecule, to trigger and regulate synaptic differentiation [96]. MAGI1 was also shown to interact with the potassium channels 4.1 (Kir4.1) acting as a scaffold protein in renal cell regulation and electrolyte homeostasis [97]. MAGI1 also interacts with soybean lipoxygenase 1 (SLO1), the pore-forming subunit of large conductance Ca^2+^-activated K^+^ (BK_Ca_) channels [98]. In mice, MAGI1 interacts with nephrin in regulating Rap1 activation in podocytes, necessary to maintain long term slit diaphragm structure. MAGI1 knock-out (KO) mice show normal glomerular histology and function into adulthood, suggesting that MAGI1 is not essential for glomerular function or its loss may be compensated by other MAGI family members [99]. MAGI2 also interacts with the nephrin complex and is essential for the maintenance of the functional structure of the slit diaphragm and podocyte survival [100,101,102]. MAGI2 KO mice, unlike MAGI1, cause neonatal lethality with observable kidney failure [100,101], while podocyte-specific MAGI2 deficiency promotes glomerulosclerosis several weeks after birth [103]. In addition, reduced *MAGI2* expression or mutations have been associated with glomerular disorders such as nephrotic syndrome in mice and humans [104,105,106]. Thus MAGI2, rather than MAGI1, is essential for proper glomerular structure and function (for a broader overview on MAGI2 role in different glomerulopathies see [107]).

MAGI2 is highly expressed in the brain, where it is localized at both excitatory and inhibitory synapses [14,108]. Mice lacking MAGI2 exhibit abnormal elongation of dendritic spines, highlighting its important role during morphogenesis of neurons [109]. In the hippocampus, MAGI1 binds and regulates the GLT1 surface expression and the levels of glutamate [110]. MAGI1 also regulates the traffic of corticotropin-releasing factor receptor 1 (CRFR1), a receptor implicated in some psychiatric disorders [111]. Several studies in non-mammalian organisms (*Drosophila melanogaster*, *Caenorhabditis elegans*, or in Zebrafish) suggest an association of MAGI1 in the brain with memory and learning, but the mechanisms are still unknown [111]. Genome-wide association studies (GWAS) for the uncovering of polygenic association with depressive disorders found that *MAGI1* copy number variations and polymorphism are associated with schizophrenia, bipolar disorders, and depressive episodes [112,113,114]. The functional or causal contribution of MAGI1 to these psychiatric conditions, however, remains to be demonstrated.

### 1.6. Regulation of MAGI1 Expression

Genomic mutations, gene rearrangements, promoter methylation, and microRNAs (miRNAs) have been described to alter the expression of MAGI subfamily members in different cell types. MAGI1-specific mechanisms that regulate its expression, and in particular cause or contribute to its downregulation in cancer cells, remain largely to be elucidated. With the aim to identify novel somatic events involved in the progression of prostate cancer (PCa), Berger et al. sequenced PCa samples and found genetic rearrangements disrupting both *PTEN* and *MAGI2.* These rearrangements included two independent inversions and two long-range intrachromosomal inversions [115]. In addition to PCa, *MAGI2* rearrangements with frame shift have been also identified in malignant melanoma [116,117], another cancer type in which PTEN loss is prevalent [118]. A study on BC showed that suppression of *MAGI2* expression by miR-101 (through translational repression instead of mRNA degradation) reduces PTEN activity and AKT activation [119], while in lung adenocarcinoma cells, miR-134, miR-487b, and miR-655 were found to target *MAGI2*, leading to loss of PTEN stability [55]. MicroRNAs (miRNAs) are key post-transcriptional repressors of gene expression by base-pairing to mRNAs (resulting in mRNA degradation and impaired translation), thereby regulating a broad range of biological processes, including proliferation, differentiation, apoptosis [120] and metastasis [121]. Dysregulation of *MAGI2* expression either via somatic genomic events or post-transcriptional regulation (by micro-RNAs) appears to affect PTEN activity [119] in several cancer subtypes. In another study on PCa, *MAGI2* mRNA gene expression levels were found significantly downregulated in a cohort of clinical PCa samples and cell lines, however, in this case, the downregulation of *MAGI2* was not correlated to PTEN mRNA expression, suggesting that different genomic events caused distinct chromosomal aberrations in each gene [122]. *MAGI2* was also found to be epigenetically silenced in cervical cancer [123]. DNA sequencing studies of BC genomes have revealed additional mutations and gene rearrangements involving *MAGI3*. A study performed on whole-exome sequences of DNA from human BC tissues identified a recurrent *MAGI3-AKT3* fusion enriched in triple-negative BC lacking estrogen and progesterone receptors and *ERBB2* expression [124]. The rearrangement produces an in-frame fusion gene with a predicted MAGI3-AKT3 fusion protein that combines MAGI3 lacking the second PDZ domain, required for PTEN’s inhibitory effect on the PI3K pathway [15], together with an AKT3 region that retains an intact kinase domain. This MAGI3-AKT3 fusion leads to constitutive activation of AKT kinase [124].

Regarding MAGI1, a recent report revealed that *MAGI1* mediates tumor metastasis through the c-Myb/miR-520h/MAGI1 signaling pathway in renal cell carcinoma [125]. Somatic genomic alterations of *MAGI1* genes were also found to be associated with metachronous gastric cancer development [126], and in a DNA methylation analysis of anaplastic thyroid cancer (ATC), *MAGI1* displayed promoter methylation as well as decreased expression in ATC compared with normal thyroid tissue [127]. Aberrant DNA methylation can result in both upregulation of oncogenes and the silencing of tumor suppressor genes [128]. In this context, *MAGI1* was reported among genes aberrantly methylated in acute lymphoblastic leukemia [129]. Next generation DNA sequencing studies performed in atypical choroid plexus papilloma, a rare benign tumor of the central nervous system that is usually confined to the cerebral ventricles, identified, along other genes, missense mutations in the *MAGI1* gene [130]. Similarly, *MAGI1* mutations were found in in parathyroid carcinoma, a rare endocrine malignancy [131], and in hidradenoma papilliferum, a benign tumor of the anogenital region that harbor mutations in major driver genes of the PI3K/AKT/MAPK-signaling pathways [132]. As already mentioned, *MAGI1* copy number variations were found associated with bipolar affective disorder and schizophrenia [112]. In addition, expression differences of *MAGI1* in bipolar cases versus control subjects have also been reported [133], findings that support that MAGI1 might also be important for brain function in psychiatric disorders.

MAGI1 also seems to be regulated by other genomic-independent mechanisms. During apoptosis induced by Fas/Fas ligand, staurosporine, and UV irradiation in different models [134], MAGI1 is cleaved by caspases at Asp761 into amino- and carboxyl-terminal cleavage products. This amino-terminal fragment dissociates from the cell membrane [135], thereby weakening protein interactions at the cell–cell interface and facilitating cell–cell dissociation. Thus, MAGI1 cleavage appears to be an important step in the dismantling of cell junctions during apoptosis [134].

MAGI1 is highly expressed in ER^+^/HER2^−^ BC, and it was shown to be upregulated by estrogen/ER and necessary for estrogen/ER signaling [61]. An important question emerging from these observations, concerning the mechanisms of downregulation of MAGI1 expression. In ER^+^ BC, *MAGI1* mutations or copy number alterations are rare (observed in less than 3% of all BC, and 2% within the ER^+^/HER2^−^ BC dataset), indicating that in this particular case, DNA alterations are not the major cause of MAGI1 downregulation in this BC subtype. Reduced MAGI1 expression in ER^+^/HER2^−^ BC correlates with inflammation, based both on human transcriptomic and experimental data. Furthermore, MAGI1 was shown to be downregulated by the prostaglandin E_2_ (PGE_2_)/Cyclooxygenase-2 (COX-2) axis. Experimentally, the inflammatory mediator PGE_2_ downregulates MAGI1 in MCF7 cells, while COX-2 inhibition (with the selective COX-2 inhibitor NS-398) upregulates its expression [61]. Chronic inflammation and PGE_2_ play a critical role in cancer development and progression [136]. COX-2, the rate-limiting enzyme in PGE_2_ synthesis, is overexpressed in several cancers including BC (reviewed in [137]), and PGE_2_ exerts diverse effects on cell proliferation, apoptosis, angiogenesis, inflammation, and immunosuppression [136]. A significant increase in COX-2 and PGE_2_ are also observed during colorectal carcinogenesis [138]). In CRC, similarly to BC, MAGI1 expression is negatively regulated by PGE_2_ and positively by COX-2 inhibitors (COXIBs) [60]. The selective COXIBs celecoxib and NS-398, and the pan-COX-1/2 inhibitor ibuprofen were shown to increase MAGI1 protein (and mRNA) levels in different human CRC-derived cells. Importantly, celecoxib treatment in vivo increased MAGI1 levels and significantly reduced tumor growth, consistent with the findings in other tumor models (reviewed in [139]).

In glioma cells, the mechanisms that contribute to MAGI3 downregulation remains unknown. COX-2 was shown to correlate with glioma grade and is a negative prognostic marker [140]. The use of celecoxib effectively suppresses tumor growth and induces apoptosis in human glioma cells [141]. Given the similarity in structure, subcellular localization, and function between MAGI1 and MAGI3, it is tempting to speculate that MAGI3 may also be downregulated by PGE_2_ [59], however, this assumption needs further evaluation.

## 2. Conclusions and Open Questions

MAGI1 is emerging as a protein exerting multiple, essential cellular functions and implicated in important human pathologies, particularly in cancer and atherosclerosis. In this review, we have highlighted the current knowledge on MAGI1 functions in epithelial, endothelial, and cancer cells (summarized in Figure 2). Although progress has been made in understanding the role of MAGI1 as tumor suppressor or regulator of vascular functions, there are still many open questions, particularly regarding its regulation in different tissues, and its clinical relevance.

### 2.1. MAGI1 as Tumor Suppressor

One of the outstanding questions relative to MAGI1 function is how it exerts its function as tumor suppressor in different cancers. MAGI1 appears to modulate and/or suppress several oncogenic pathways, in particular PI3K/AKT via PTEN, the WNT/β-catenin pathway by recruiting β-catenin to E-cadherin complexes, and the MAPK/ERK signaling pathway. It also suppresses FAK phosphorylation and decreases YAP activity. Structurally, MAGI1-c contains an additional carboxyl-terminal 251 amino acids sequence forming a NLS targeting it to the nucleus [3], suggesting a possible direct role in the regulation of gene transcription. MAGI1 was shown to be essential for NF-κB activation in ECs [81], a key transcription factor that controls the expression of many essential pro-inflammatory and survival genes [142]. However, to date there is not a clear comprehensive view of different genes regulated by MAGI1. It is likely that more oncogenic pathways are modulated by MAGI1 and that different pathways may be differentially affected depending on the context and the cell type.

### 2.2. MAGI1 Link with Inflammation

Inflammation is considered a hallmark of cancer [143], and increasing evidence indicates that it may promote acquired endocrine resistance and more aggressive progression of ER^+^ BC. About 75% of all BCs express ERα at the time of diagnosis [144], and there is a strong evidence that estrogen plays a critical role in the development and progression of the disease [145]. ERα-positive BCs rely on estrogen signaling for proliferation. Thus, the most effective strategy of these hormone-sensitive tumors is to block estrogen action using endocrine therapy (tamoxifen or aromatase inhibitors). Unfortunately, de novo or acquired resistance towards these agents has become a major clinical obstacle [144]. MAGI1 is highly expressed in ER^+^BC and its expression is upregulated by estrogen and negatively correlates with inflammation in ER^+^/HER2^−^ BC patients. Loss of MAGI1 in ER^+^ BC could be relevant in the acquired resistance to hormonal therapy in this particular subtype of BC. Within this subtype, low MAGI1 levels predict for a more aggressive behavior, while high levels correlate with a lower risk of relapse in ER^+^/HER2^−^ patients, especially in those treated with tamoxifen compared to untreated or chemotherapy-treated patients (reviewed in [61]). The use of NSAIDs increases MAGI1 expression in both, BC and CRC [60,61]. Preclinical and clinical studies have demonstrated that NSAIDs have chemopreventive effects on BC [146]. The biological effects of NSAIDs could be particularly relevant to the perioperative period, marked by an activation of inflammatory pathways [147]. Animal models and retrospective studies showed that perioperative administration of NSAIDs has been associated with lower risk of cancer recurrence [148,149]; however, other studies do not support their use [150,151]. Regular use of NSAIDs or COXIBs were also described to reduce the risk of other cancers, such as lung, esophageal, or CRC [152]. However, and similarly as in BC, their prolonged use is associated with unacceptable gastrointestinal and cardiovascular side effects [153]. Anecdotally, MAGI1 was identified in a screen for NSAIDs-induced molecules to target as an alternative to the long-term use of NSAIDs as tumor preventive agents [60]. It is important to unravel the signaling pathways and mechanisms that induce MAGI1 loss in different cancer types. PGE_2_ was shown to suppress MAGI1 expression in cancer cells [60,61]. A better understanding of the mechanism and/or the signaling pathways that induce MAGI1 loss could enable the identification of novel and safer potential therapeutic targets, with the purpose of maintaining high MAGI1 levels under inflammatory conditions, as an alternative to NSAIDs or COXIBs-based treatments. One way could be testing whether inflammatory cytokines, such as IL1, IL6, or TNFα may also decrease MAGI1 protein levels, thus becoming potential targets.

Different MAGI subfamily members have been described to play a negative role in inflammation in other human diseases. Many studies have linked MAGI1 to different chronic inflammatory diseases, such as inflammatory bowel disease (IBD) [154], psoriasis [155], Crohn’s disease [156], celiac disease [157], and microscopic colitis (MC), a multifactorial condition characterized by chronic inflammation of the colon and associated with increased paracellular permeability [158]. Decreased expression of MAGI3 has also been described to contribute to the pathogenesis of IBD [159], and genetic variations of the gene *MAGI2* have similarly been described in Crohn’s disease, ulcerative colitis, and celiac disease [160,161]. MAGI1, together with other molecules, plays a pivotal role for the barrier function of epithelial tight junctions, ensuring a highly selective barrier permeability, acting against commensal bacteria and foreign antigens. Consequently, the disruption of its homeostasis may have dramatic effects on the mucosal integrity, which has been shown to contribute to the development of different gastrointestinal diseases (reviewed in [156]).

### 2.3. Regulation of Endothelial Function

While MAGI1 loss seems a common feature in many different cancers and inflammatory diseases, ECs activation and consequent MAGI1 upregulation contribute to the pathogenesis of atherosclerosis [81]. A better understanding of the specific signaling events involved in the mechanism of EC activation is key for the development of therapeutic strategies against vascular diseases. Fluid shear stress was shown to induce MAGI1 expression in ECs, which in turn regulates eNOS phosphorylation and NO production [77]. As abnormalities in vascular NO production and transport results in EC dysfunction, leading to various cardiovascular pathologies [80], it will be interesting to evaluate the role of MAGI1, a modulator of NO, in cardiovascular diseases such as hypertension, other angiogenesis-associated disorders or in diabetes, where perturbed expression of NO was reported. In addition, and in the context of MAGI1 regulating the permeability of ECs, it would be of interest to evaluate in further detail the implication of MAGI1 in leukocyte extravasation, an important process that takes place during vascular damage or inflammation processes and that involves a variety of cell adhesion molecules that constitute the cell junctions.

MAGI1 colocalizes to mature focal adhesions in ECs and regulates integrin-dependent cell adhesion and migration [76]. Many questions arise concerning this point, like how MAGI1 is recruited to focal adhesions; how it regulates focal adhesion maturation; and which are the binding partners that interact with MAGI1 in the adhesiome. Considering the essential role of integrins in endothelial cellular functions, addressing these questions may uncover yet unknown aspects of integrin function regulation in vascular biology.

### 2.4. Clinical Significance

While it is clear that MAGI1 expression is reduced during cancer progression, it is not clear yet whether MAGI1 levels may be used to stratify cancer patients for prognostic purposes and, more importantly, for therapeutic decisions. As mentioned before, MAGI1 could serve as a prognostic marker for the identification of patients at high-risk of relapse in ER^+^ BC [61]. MAGI1 is a favorable prognostic marker in renal cancer [162], while in HCC [50] and glioma [58], low MAGI1 expression is associated with poor prognosis. MAGI2 was proposed as a novel diagnostic marker [122] and suggested as a predictor of tumor recurrence in prostate cancer [163]. In addition, it may eventually help in patient stratification for clinical trials using PI3K pathway inhibitors [115]. Identifying putative biomarkers in cancer is crucial, and might be useful for the selection of patients who are the most likely to benefit from a specific therapeutic approach [164].

### 2.5. Therapeutic Implications

Could MAGI1 become a direct therapeutic target to treat human pathologies? In conditions where MAGI1 expression or function is decreased, this may not be an option, as it would require increasing protein expression. Should we, however, learn how to pharmacologically induce MAGI1 expression? MAGI1 may become a “reverse target” or a rational for considering such drugs. To this regard, the observed induction of MAGI1 expression by NSAIDs is consistent with the tumor suppressive capacities of both MAGI1 and NSAIDs. In the context of vascular diseases, and in view of the essential roles of MAGI1 in maintaining basic cellular and tissue functions (i.e., regulation of cell–cell and cell–matrix adhesions, stabilization of epithelial and endothelial integrity, mediating podocyte and synaptic functions), systemic pharmacological MAGI1 inhibition might not represent a feasible option, as it will be overshadowed by massive and generalize unwanted effects. In addition, due to its important role in regulating permeability, MAGI1 inhibition may cause increased EC permeability, which may enhance atherosclerosis formation [165].

In conclusion, more experimental studies at molecular, cellular, and tissue levels in different contexts and cell types will be necessary in order to fully understand MAGI’s roles, regulation, prognostic value, and therapeutic relevance. We believe that additional information about MAGI1 will be revealed in the upcoming years, and it will be important not only to look at additional cancer fields or cell types, but also to re-evaluate existing studies about MAGI1 with greater attention—and maybe a different perspective—to fully understand MAGI1 implication and significance in both, cancer progression and vascular biology.

## Figures and Tables

**Figure 1 cells-10-01494-f001:**
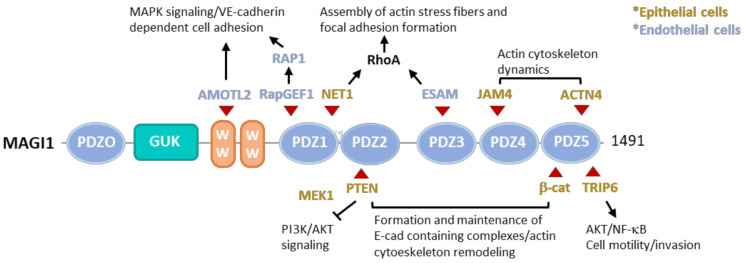
Schematic representation of MAGI1 domain structure and selected interacting molecules relevant for cancer and vascular biology. In endothelial cells, AMOTL2 is needed for the connection of the VE-cadherin/catenin complex to MAGI1 and actin filaments, and for transmission of forces between neighboring cells [42]. MAGI1 interacts with RapGEF1, activating Rap1, which regulates the MAPK signaling pathway and VE-cadherin-mediated cell adhesion [43]. ESAM binds MAGI1 and recruits it to cell–cell contacts [21], promoting and strengthening mature cell–cell adhesion via activation of RhoA [44]. In epithelial cells, MAGI1 interacts with NET1 [22], which in turn activates RhoA [45], leading to the assembly of actin stress fibers and focal adhesion formation [46]. MAGI1 colocalizes with E-cadherin, β-catenin, and PTEN, regulating the formation and maintenance of E-cadherin containing complexes [25,26]. MEK1 is necessary for PTEN membrane recruitment [28]. TRIP6 competes with β-catenin for the binding to the MAGI1/PTEN signalosome, destabilizing E-cadherin complexes and promoting cell motility through the regulation of AKT/NF-κB targets [27]. At tight junctions, MAGI1 colocalizes with JAM4 [20] and α-actinin-4 [31], regulating the actin cytoskeleton dynamics. Arrows (↑) indicate activation/induction, blunt ended lines (T) indicate inhibition/blockade. RapGEP: Rap guanine nucleotide exchange factor 1; Rap1: Ras-related protein 1; MAPK: mitogen-activated protein kinase; VE-cadherin: vascular endothelial-cadherin; NET1: neuroepithelial cell transforming 1; RhoA: Ras homolog family member A; ESAM: endothelial cell adhesion molecule; AMOTL2: angiomotin like 2; PTEN: phosphatase and tensin homolog; PI3K/AKT: phosphoinositide 3-kinase/protein kinase B; TRIP6: thyroid hormone receptor interactor 6; NF-κB: nuclear factor kappa B; ACTN4: α-actinin-4.

**Figure 2 cells-10-01494-f002:**
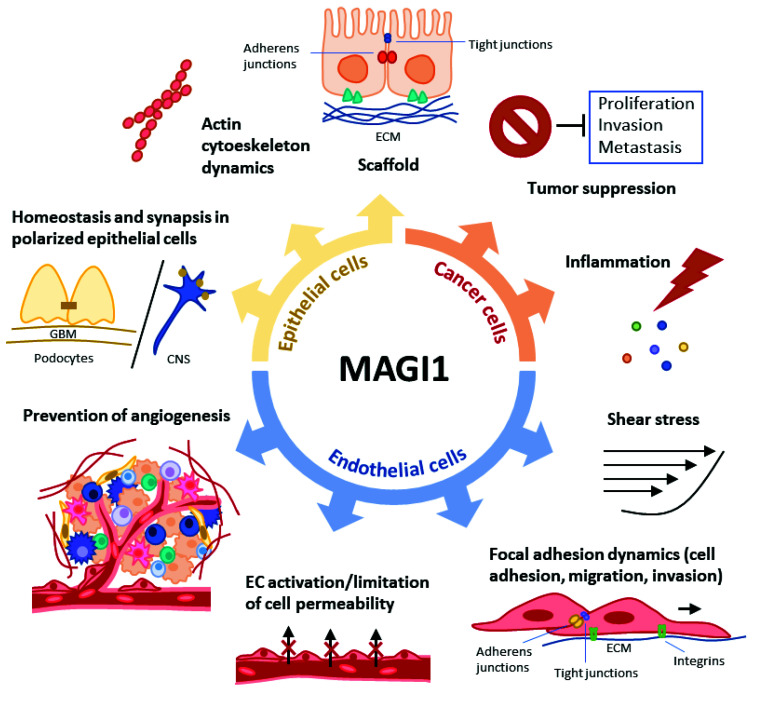
Summary of MAGI1 functions in epithelial, endothelial (EC), and cancer cells. In epithelial cells, MAGI1 is present in cell–cell contacts such as tight and adherens junctions and acts as a scaffold molecule. MAGI1 is also involved in regulating the actin cytoskeleton dynamics. In polarized epithelial cells, it is involved in receptor synaptic localization and homeostasis of ion channels. In ECs, MAGI1 is induced in response to fluid shear stress. MAGI1 also controls focal adhesion dynamics (stabilizing cell adhesion and decreasing migration and invasion). MAGI1 regulates EC activation and reduces cell permeability and angiogenesis. In cancer cells, MAGI1 acts as tumor suppressor preventing proliferation, invasion, and metastasis. There is increasing evidence of a link between MAGI1 and inflammation. ECM extracellular matrix; GBM glomerular basement membrane; CNS central nervous system.

**Table 1 cells-10-01494-t001:** Summary of the tumor suppressor functions of MAGI in different cancers.

Subfamily Member	Function	Mechanism	Model	Reference
MAGI1-b	Tumor suppressor role	Ectopic expression potentiates interaction with PTEN, promotes cell–cell aggregation, reverts Src-induced invasiveness, decreases AKT activity	Kidney epithelial MDCKts-src cells	Kotelevets et al., 2001
MAGI1	Tumor suppressor in hepatocellular carcinoma (HCC)	MAGI1 transfection inhibits cell migration and invasion by upregulating PTEN	HepG2 cells	Zhang et al., 2011
		Decreased expression associated with poor prognosis in HCC patients, and its downregulation correlates with vascular invasion of HCC tissues	HCC datasets	Zhang et al., 2012
MAGI2	Tumor suppressor in hepatocellular carcinoma (HCC)	Inhibits cell migration and proliferation through downregulation of p-FAK and p-AKT	HCC cell lines	Hu et al., 2007
MAGI1	Tumor suppressor in adult T-cell leukemia (ATL)	Inhibits AKT activity through PTEN and MEK1. Knockdown increases AKT and MEK activity. Upregulation reduces cell growth	T-cells	Kozakai et al., 2018
MAGI3/ MAGI1	Tumor suppressor in glioma	Overexpression of MAGI3 upregulates PTEN, inhibits phosphorylation of AKT, and suppresses cell proliferation	Glioma cell lines	Ma et al., 2015
		MAGI3 regulates negatively Wnt/β-catenin signaling. Knockdown enhances cell proliferation in vitro; overexpression suppresses tumor growth in vivo; associated negatively with tumor grade and poor prognosis	Glioma cell lines; in vivo studies; glioma datasets	Ma et al., 2015
		MAGI1 enhances proliferation and inhibits apoptosis, increases the Wnt/β-catenin signaling pathway, p-AKT and reduces E-cadherin and PTEN expression. Overexpression of MAGI1 inhibits tumor growth in vivo	Glioma cell lines; in vivo studies	Lu et al., 2019
		MAGI1 overexpression inhibits proliferation, migration and invasion of glioma cells by regulating EMT through AKT, MMP2, MMP9, and the E-cadherin/*N*-cadherin/vimentin pathway	Glioma cell lines	Li et al., 2019
MAGI2	Tumor suppressor in lung adenocarcinoma	Regulated by TGF-β1. TGF-β1 stimulation induces overexpression of miR-134/487b/655, that silence MAGI2	Lung adenocarcinoma cell lines	Kitamura et al., 2014
MAGI1	Tumor suppressor in colorectal cancer (CRC)	Negative regulator of Wnt/β-catenin signaling. Silencing enhances Wnt signaling, migration, and invasion in vitro; overexpression attenuates tumor growth and metastasis in vivo by inhibiting Wnt signaling	CRC cancer cell lines; in vivo studies	Zaric et al., 2012
MAGI1	Tumor suppressor in ER^+^/HER2^−^ breast cancer (BC)	High expression associated with better prognosis/low with higher histological grade, aggressive phenotype, and worse prognosis. Downregulation promotes cell proliferation and survival through activation of PI3K and Wnt signaling, and enhances tumor growth and metastasis in vivo	BC cell lines; in vivo studies; BC datasets.	Alday-Parejo et al., 2020
		Loss of MAGI1 leads to accumulation of E-cadherin and AMOTL2 in the cells, which increases stiffness, decreases YAP activity, and increases ROCK and p38 stress pathways	BC cell lines; in vivo studies	Kantar et al., 2021
MAGI1	Tumor suppressor in gastric cancer	Downregulation enhances migration and invasion by altering the expression of MMPs and EMT-related molecules via inhibiting the MAPK/ERK signaling pathway	Gastric cancer cell lines	Jia et al., 2017
MAGI1	Tumor suppressor in cervical cancer	Proteolytic substrate of the HPV-16 and HPV-18 E6 oncoproteins. MAGI1 restoration induces apoptosis and represses cell proliferation	Cervical cancer cell lines	Kranjec et al., 2014
MAGI2	Tumor suppressor in prostate cancer	Its mutation is theorized to contribute to prostate carcinogenesis by driving AKT phosphorylation and altering PI3K signaling, thus disrupting PTEN signaling		Brenner & Chinnaiyan, 2011

PTEN: Phosphatase and tensin homolog; src: sarcoma; AKT: protein kinase B; HCC: hepatocellular carcinoma; FAK: focal adhesion kinase; MEK1: MAPK/ERK kinase 1; Wnt: wingless; EMT: epithelial–mesenchymal transition; MMP: matrix metalloproteinase; TGF-β1: transforming growth factor beta 1; CRC: colorectal cancer; BC: breast cancer; PI3K: phosphoinositide 3-kinase; AMOTL2: angiomotin like 2; YAP: yes-associated protein 1; ROCK: Rho-associated protein kinase; MAPK/ERK: mitogen-activated protein kinase/extracellular signal-regulated kinase; HPV: human papillomavirus.

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
