# Peer review of "MAGI1, a Scaffold Protein with Tumor Suppressive and Vascular Functions"

_cells, 2021, doi:10.3390/cells10061494_

Round 1
Reviewer 1 Report
The authors, Janine Worthmuller and Curzio Ruegg, do a nice job summarizing the overall field of MAGI1 biology. They subdivide the review into sections that organize the article well, and they do a good job reviewing the diverse roles of MAGI proteins. The two figures and table are excellent and overall very helpful.
I find the cancer section to be quite comprehensive and without any additional necessary changes.
I would add additional text and references to the epithelial portion, in particular the podocyte section. It is clear in podocytes that MAGI2 and MAGI1 function are non-redundant. MAGI2 KO mice (three distinct mouse models - (AM J Path 2014, PNAS 2014, JASN 2017 ) develop severe podocyte injury with progression to ESRD while MAGI1 KO mice have only a very subtle podocyte phenotype (JBC 2016). This is recapitulated in human congenital nephrotic syndrome cases that are caused by MAGI2 mutations (Nat Comm 2018, JASN 2017). It is clear that in podocytes MAGI2 is the dominant MAGI isoform. This does not appear to be the case in cancer biology, where both MAGI1 and 2 seem to have important roles although animal data for the role of these proteins in cancer is much more limited.
Reviewer 2 Report
Comments to the Author
The authors conducted this review well, and the methods used are appropriate. The review presented clearly, and the manuscript was well written. The review suggested that the MAG1 as Scaffold Protein with Tumor Suppressive and Vascular Regulatory Functions. To improve the article, the authors need to consider the following suggestions.
1.The authors need to discuss how the MAG-1 major role in vascular biology. Currently they explained very minimal about MAG1 role in the VB.
2.The authors need to describe how they MAG1 role in EC- Leukocyte/ monocyte interactions.
3.The authors need to discuss about the implication of MAG1 role in angiogenesis and viral infection models.
